# Laboratory Study of the Effects of the Mixer Type and Mixing Time on the Volumetric Properties and Performance of a HMA with 30 Percent Reclaimed Asphalt Pavement

**DOI:** 10.3390/ma16031300

**Published:** 2023-02-03

**Authors:** Marc-André Bérubé, Sébastien Lamothe, Kevin Bilodeau, Alan Carter

**Affiliations:** 1Construction Engineering Department, École de Technologie Supérieure (ÉTS), Montreal, QC H3C 1K3, Canada; 2BauVal Inc. (Le Groupe), Boucherville, QC J4B 6T3, Canada

**Keywords:** mixer type, mixing time, hot-mix asphalt, reclaimed asphalt pavement, semi-open graded mix, ITSM, IDT, CT_index_, cracking resistance, volumetric properties

## Abstract

This study examined the effects of the laboratory mixer type and mixing time on a hot-mix asphalt (HMA) using three different types of mixers and four different mixing times. The asphalt mix used is a semi-open graded mix (ESG-10) with 30% reclaimed asphalt pavement (RAP), and a range of tests were conducted including bitumen extraction by ignition, particle size distribution, maximum specific gravity (G_mm_), a SUPERPAVE gyratory compactor (SGC), bulk specific gravity (G_mb_), indirect tensile stiffness modulus (ITSM), and indirect tensile strength (IDT). The statistical analysis of variance (ANOVA) was also applied to quantify the effect of mixer type and mixing time. The results indicated that both mixing type and time had a significant effect on the properties of the HMA (volumetric properties and compactability) and that the type of mixer used also affected the performance of the HMA (stiffness and cracking resistance), with some mixers producing asphalt mixes with better properties than others. The study ultimately demonstrated that it is possible to produce a mix that exhibits good performance and meets or does not meet the compactability specifications depending on the mixer type used.

## 1. Introduction

The use of reclaimed asphalt pavement (RAP) in new hot-mix asphalt (HMA) has become more common in recent years due to its potential to reduce waste and improve the sustainability of asphalt production [1]. However, it is important to understand how RAP can affect the properties and performance of the resulting mix [2,3]. The inclusion of RAP in new asphalt mixes can have several effects, both positive and negative. One potential benefit of using RAP is cost saving, as it is typically cheaper than using all-new materials. RAP can also provide environmental benefits by reducing the demand for new aggregate and bitumen materials, which can help conserve natural resources. In addition, RAP can improve the strength and durability of the asphalt mix. Despite those potential benefits of RAP, it is crucial to carefully evaluate the potential effects of adding RAP before including it in the asphalt mix, particularly its performance at low temperatures [4]. The behavior of the asphalt mix may become more fragile at low temperatures when RAP is used [5].

Studies on laboratory-produced mixtures have found that adding low amounts of RAP (15–20%) does not significantly affect the stiffness and strength of the mix at different temperatures [6,7,8]. However, increasing the RAP content beyond 20% improves the mixture’s stiffness and strength, resulting in increased resistance to rutting [9]. For higher RAP contents (>40%), the stiffness of the mix increases significantly at various temperatures when no changes to the virgin binder grade are made [10,11]. The majority of Canadian agencies have established a range for the maximum RAP content allowed in new mixes, with the typical range being between 20% and 50%. The specific maximum RAP content in new mixes can vary depending on the province or territory, but on average, the RAP content used in new mixes is around 10% [12].

The evaluation of HMA typically involves a range of tests to assess the performance and quality of the mix. One common test is the indirect tensile strength (IDT) test, which measures the resistance of the asphalt mix to deform under indirect tension. This test can provide valuable information about the tensile strength and fatigue resistance of the mix, which are important factors in determining its performance in the field [13]. From those data, it is possible to calculate the crack tolerance index (CT_index_), which is a measure of the ability of asphalt pavement to resist the formation of cracks by considering the failure energy (G_f_). Higher CT_index_ values indicate greater crack resistance and improved performance of the asphalt pavement. The CT_index_ is used to predict the crack formation in asphalt pavement and to inform the design and construction of the pavement [14]. G_f_ can be used to predict the behavior of asphalt pavement under various loading conditions and to design asphalt mixes that are suitable for different applications [15,16]. It is only one factor that needs to be considered when designing and evaluating asphalt pavement. Another commonly used test for HMA is the indirect tensile stiffness modulus (ITSM) test, which measures the stiffness of the asphalt mix. This test can provide valuable information about the elasticity of the mix, which is important for predicting its behavior under different loading and temperature conditions [17].

In addition to IDT, ITSM, and CT_index_, the compactability of the HMA mix can be evaluated using a shear gyratory compactor (SGC) test. This test involves compacting the mix to a given bulk specific gravity (G_mb_) [18]. This information can be used to assess the suitability of the mix for different applications and to optimize the compaction process in the field. Finally, the volumetric properties of the HMA, such as air voids (V_a_) and effective bitumen content (V_be_), can also be evaluated. These properties are important for predicting the mix’s performance in the field and ensuring that it meets the specified requirements [9]. The results of these tests can provide valuable insights into the quality and performance of the HMA mix and inform the development of more effective and sustainable asphalt production processes [19]. Mixing time and the type of mixer can affect the properties of the mix [20]. Increasing the mixing time by 1 to 2 min could increase the coating of the aggregates for a higher viscosity bitumen without causing any problems with mix performance [21]. There is, however, limited information available on the impact of the laboratory mixer type on the properties and performance of HMA with a high content of RAP. It could, however, be mentioned that higher mixing time or higher mixing energy could allow better reactivation of the RAP bitumen, which would result in different properties of the mixes.

The main objective of this study is to evaluate, in the laboratory, the effects of the type of mixer and the mixing time on the volumetric properties and the performance of a semi-open graded mix, commonly used for a surface/friction layer, but with a high RAP content (30%). Another goal of this study is to investigate the influence of the compactor on the air voids produced by different mixer types and mixing times by comparing the results obtained using the different methods. By investigating the effects of these factors on the mix, this study aims to provide valuable insights into the optimum mixing conditions for producing high-quality HMA with a high RAP content. This information can support the development of more effective and sustainable asphalt production processes.

## 2. Experimental Program

The experimental program described in Figure 1 used different mixers and mixing times to investigate the effects on the volumetric properties and performance (stiffness and cracking resistance) of HMA with a high content of RAP (30%). The mix design was initially developed using a Hobart (HB) tabletop mixer and a mixing time typically used for HMA (45 + 30 s) without RAP following the Quebec mix design method. This mix was selected to ensure that the resulting mix met the specifications for asphalt mixes in Quebec. The goal of the program was to evaluate the effects of the mixer type and mixing time for producing HMA with high RAP content. Detailed information about the materials, the methods, and the tests are presented in Section 3.

## 3. Materials, Methods, and Tests Selected

### 3.1. Materials

In this study, high resistance to abrasion, friability, and impact (MD and LA values) crushed granite virgin aggregates were used with a non-modified virgin bitumen (PG 52S-34), and RAP material from a single stockpile. The properties of the virgin aggregates are shown in Table 1. As is often the case, we do not have specific information on the type of RAP aggregate.

The virgin bitumen and the RAP bitumen were characterized to determine their high (H) and low (L) temperature grading, as well as the appropriate mixing and compaction temperatures. The results of this characterization are shown in Table 2. The S value (the n value) of the virgin bitumen grade indicates a standard traffic-level resistance. It is important to mention that a soft, unmodified asphalt binder was selected to compensate for the hard RAP asphalt and to ensure that the combined bitumen would perform well at low temperatures.

Using the results obtained (Table 2), a blending chart was used to determine the mixing and compaction temperatures for the HMA. The higher viscosity of the RAP bitumen necessitated the setting of the mixing temperature (T_mix_) at 145 °C and the compaction temperature (Tc_omp_) at 135 °C. These temperatures were selected to ensure the optimal coating of aggregates and the performance of the HMA in the study. The properties of the RAP are shown in Table 3.

### 3.2. Preparations of Raw Materials, Mix Design, and HMA Selection

The virgin aggregates were sieved, washed, and weighed individually. The RAP material was dried in an oven at 40 °C for 3 days to take out the water without oxidizing it and then mechanically separated (aggregates splitter) following the LC 21-015 Quebec test method and a 30% portion by the total weight of aggregate was used for the mix. This is a common practice in the industry [22,23]. This resulted in a semi-open graded mix (ESG-10 mix), commonly used for surface/friction layer in the province of Quebec, with a nominal maximum aggregate size (NMAS) of 10 mm and a high content of RAP material (30 percent). The particle size distribution of the targeted mix is shown in Figure 2. The total bitumen content (or asphalt content: AC) of the mix is 5.6 percent of the total weight of the mixture. By taking into consideration that 100 percent of the RAP bitumen is mobilizable, a 4.1% bitumen addition (PG 52S-34) was used. The recycled binder ratio (RBR) corresponds to 27%, and this value was used to obtain the T_mix_ and T_comp_ in the blending chart (not shown here). The materials were carefully selected and prepared to ensure the quality and consistency of the resulting asphalt mix. This is essential for achieving accurate and reliable results from the study.

### 3.3. Mixer Types and Mixing Times

Three (3) different types of mixers were used to produce the HMA, as shown in Figure 3. The first type of mixer was a countertop mixer (HB: Hobart model A200), which is commonly used in laboratory settings. The second type of mixer was a laboratory pugmill twin-shaft mixer (PM: Wirtgen model WLM 30), which is typically used for producing cold mixes in a laboratory. This type of mixer is similar to those used in asphalt plants to produce HMA, which means that the mix produced by this mixer may be more representative of the mixes typically used in the field. The third type of mixer used in the study was a thermoregulated mixer (TR: InfraTest model 30 l), which is a temperature and speed-controlled mixer. This type of mixer allows for precise control over the mixing conditions, which is important for producing high-quality asphalt mixes.

The mixing time for each mixer type was selected based on the mixing time typically used for mixes without RAP in accordance with the LC 26-003 and LC 26-004 Quebec test methods. The first (1st) mixing time was the mixing time used for mixes with no RAP, and the second (2nd) mixing time was doubled for all subsequent mixes in order to cover a wide range of mixing times.

Table 4 shows the mixer types and mixing times (1st and 2nd) used in this study. With the combined mixer types (×3) and mixing times (×4), a total of twelve mixes (Table 4) with the same reference material were made in our laboratory.

### 3.4. Conditioning of Materials and Mixing Conditions

The conditioning of the raw materials depended on the type of mixer (Table 4). The aggregates for all mixers were heated in a draft oven the night before mixing. For the TR mixer, all components were at the T_mix_ (145 °C). For the HB and PM mixers, considering the loss of temperature during the mixing process, the aggregates were overheated at 185 °C to reach T_mix_ when combined with the RAP. To limit the oxidation of the RAP, the pre-dried RAP for all mixers was conditioned at 60 °C the night before mixing and then added in the HB and PM mixer at this temperature. The TR mixer was heated like the aggregates to the T_mix_ (145 °C) before mixing. Virgin bitumen was heated at 145 °C three (3) hours prior to mixing, and its temperature was checked. The temperature was monitored throughout the mixing process with an infrared thermometer.

### 3.5. Batch Size, Splitting, and Conservation before testing

For the TR and PM mixers, a single (1) batch of 20 kg was prepared. For the HB mixer, two (2) smaller batches of 10 kg each were combined to create a total of 20 kg. The resulting material was then mechanically divided into two (2) equal parts in accordance with the LC 21-010 Quebec test method. These portions were placed in two cardboard boxes and stored for seven (7) days before being reheated. The mixes were reheated at 115 °C and divided again to obtain the necessary weight for testing the volumetric properties and performance of the mixes under different mixing conditions.

### 3.6. Determining HMA Volumetric Properties and Compactability

The maximum specific gravity (G_mm_) of the asphalt mixes was determined using the LC 26-045 Quebec test method, and a shear gyratory compactor (SGC) was used in accordance with the LC 26-003 Quebec test method, as depicted in Figure 4. These results were used to calculate the volumetric properties of the mixes, including the percentage of absorbed asphalt binder by mass (P_ba_), the effective volume of asphalt binder (V_be_), the voids in the mineral aggregate (VMA), and the voids filled with asphalt (VFA). These properties provide valuable information about the characteristics and performance of the asphalt mixes. The value obtained using the SGC test can be used to assess the compactability of the mixes and to determine the air void content (V_a_) in the mixes. For the SGC test, the mixes are compacted at a temperature of 135 °C (T_comp_). This information is useful for understanding the characteristics and performance of the asphalt mixes.

### 3.7. Specimen Preparation and Determining HMA Performance (Stiffness and Crack Resistance)

In order to carry out the tests characterizing the performance of the mixes, it is necessary to prepare specimens. For this purpose, a second compaction method was performed using the Marshall hammer (40 blows per side and six (6) specimens per mix with a diameter of 100 mm and a thickness of 63 ± 5 mm) according to the LC 26-020 Quebec test method for all 12 mixes. Again, the mixes were compacted at a temperature of 135 °C (Tcomp). Forty blows, instead of sixty as specified by the test method, were chosen to have a higher air void content (V_a_) for Marshall specimens and to make them as comparable as possible to SGC specimens. In particular, this value of 40 blows is not arbitrary and is normally used when the moisture resistance of the asphalt mix (specimens) is to be checked, as indicated in the Quebec test method LC 26-001. The hydrostatic weighing was used to determine the bulk specific gravity (G_mb_) and air voids (V_a_) of each specimen following the LC 26-045 Quebec test method, as shown in Figure 5. No additional aging was performed on the specimens before testing. For all performance evaluations, the statistical analysis of variance (ANOVA) was also applied to quantify the effect of mixer type and mixing time.

Of the six specimens, three (3) Marshall specimens with similar air voids (V_a_) for each mix were selected for the indirect tensile stiffness modulus (ITSM) test, which is based on the NF EN 12697-26 standard. This test was chosen because it is a non-destructive test, and the tested specimens can be kept for the indirect tensile strength (IDT) test. For the ISTM test, four (4) testing temperatures were selected for each mix (−20, −10, 0, and 10 °C). During the test, 10 cycles of compression tension were applied and the horizontal strain was recorded. The ITSM test is a method for determining the tensile strength and elastic modulus of asphalt materials.

The IDT tests were conducted according to the ASTM D6931-17 standard at 25 °C and −20 °C, with three (3) specimens tested for each temperature for a total of six (6) specimens for this test. To achieve those tests with precision, a hydraulic press with a temperature control chamber was used. Figure 6 presents the ITSM and ITD devices.

From the IDT results, it was possible to evaluate the crack tolerance index (CT_index_) following the ASTM D8225-19 standard. According to the ASTM D8225-19 standard, it is possible to calculate the work of failure (W_f_) and the failure energy (G_f_). W_f_ is a measure of the energy required to cause a material to break or fail. It is represented by the area under the curve in the IDT test. G_f_ is a measure of the amount of energy required to break a specimen, considering its dimensions (thickness and diameter). These values can be used to evaluate the performance and durability of asphalt mixes in different applications, and to optimize their use in construction and paving projects. They provide valuable information about the strength and resilience of asphalt under different conditions and can be used to select the most suitable asphalt mix for a given project.

## 4. Results and Analysis

### 4.1. HMA Volumetric Properties and Compactability

The volumetric properties evaluated are the G_mm_, the percentage of absorbed asphalt bitumen by mass (P_ba_), the effective volume of asphalt binder (V_be_), the voids in the mineral aggregate (VMA), and the voids filled with asphalt (VFA). With the SGC results at N_design_ (80 gyrations for ESG-10 mix), it was possible to evaluate the specific bulk gravity (G_mb_) of the HMA and after, to determine the V_a_, V_be_, VMA, and VFA values. Table 5 shows the volumetric properties.

The G_mm_ value of a mix can vary depending on the type of mixer used and the mixing time. In this test, the G_mm_ values ranged from 2.509 to 2.530, with the highest values occurring at the first mixing time for each mixer. The G_mm_ is affected by the amount of asphalt bitumen absorbed by the aggregates in the mix. In principle, higher absorption leads to a higher G_mm_ value [9]. The absorbed bitumen could be an indicator that the bitumen from the RAP is more activated with the mixing time and its mobilization is increased. However, it is possible that there is something else that explains this, requiring further analysis of results. In this test, the maximum acceptable difference between two tests performed by the same operator is 0.011, which is higher than the difference of all TR mixes. HB mixes have a 0.017 difference and PM mixes have 0.015, which remains low.

The Vbe value, a mix design criterion used in Quebec, is aimed at being at 12.2 ± 0.1%. For all mixes, the first mixing time has the lowest Vbe value, which increases with subsequent mixing times. For TR and HB mixes, the second mixing time (TR-2 and HB-2) has the highest Vbe value, which then stabilizes for the next mixing time. For PM mixes, Vbe continues to increase with mixing time (PM-1 to PM-4).

VMA are the air-void spaces between the aggregates, including the space filled with the bitumen of the compacted mix. For all mixes, the VMA is over 15%, indicating a good mix, and it is not really affected by the mixing time. It slightly decreases with mixing time (1 to 4). VFA is the percentage of VMA containing bitumen. VFA rise with the increase in the mixing time. The VFA value increases with mixing time, with the least impact on TR mixes and the most impact on PM mixes. It is affected by the fine particles (80 µm sieve-passing), as shown in Section 4.1.2.

#### 4.1.1. Air Voids Evaluation

Table 6 shows the air voids (V_a_) with the SGC at 10 (N_ini_), 60, 80 (N_design_), 100, 120, and 200 (N_max_) gyrations and the Quebec specifications for the ESG-10 mix. It is possible to observe in this table that the mixer type and mixing time affect the compactability of the mixes and have an impact on the mixes meeting the targeted specification. Mix design was conducted at first with the HB-1 and it respects all of the Quebec specifications. In all the other HB mixes, the voids are too low at 200 gyrations (shown in red in the table). In addition, for the HB-4 mix, the voids are too low at N_design_.

For the TR-1 and PM-1 mixes, the voids at N_design_ are too high and do not meet the Quebec specifications (Table 6). However, mixing for longer periods can help to reduce these voids. Mixing for a proper amount of time can also help to activate the RAP bitumen and improve the compactability of the mixes. In addition, longer mixing can fracture and wear the aggregates and produce finer particles that fill and reduce air voids. A strong correlation between the V_be_ and VFA in the function of the air voids at N_design_ can be observed in Figure 7. Although the variation in VFA is higher, it follows the trend with the voids obtained at N_design_.

Figure 8 shows the air voids obtained with Marshall compaction. Even though 40 blows were used instead of 60 to compact the Marshall specimens, Figure 8 shows that the voids are lower with Marshall compaction (M), around two percent lower than the air voids at N_design_ with SGC compaction, which is commonly observed in asphalt specimens.

SGC and Marshall compaction do not show the same trend, but some similarities can be observed. For all the mixes, the voids are the highest at the first mixing time and decrease with the increase in the mixing time. For the HB mixer, Marshall compaction affects the voids linearly with the mixing time, but a certain stabilization is observed in the SGC compaction. TR mixes show the same void variability for both compaction methods. For PM mixes, the effect of mixing on voids varies depending on the compaction method used. When using SGC compaction, voids decrease with increasing mixing time, with a stabilization occurring between the second and third mixing times. However, when using Marshall compaction, voids decrease with increasing mixing time for the first three mixing times, but then increase for the fourth mixing time. Finally, it is shown that the laboratory mixer type and mixing time should influence the mix proportion of the aggregates to meet the void SGC specifications.

#### 4.1.2. Bitumen Content and Gradation

Table 7 presents the asphalt content (AC) and particle size distribution of various mixes that were tested in an ignition oven. These mixes were all composed of the same aggregate size, RAP, and bitumen content, allowing for comparison between the different mixes. The previous results indicate that the virgin aggregates used in these mixes do not require a correction factor with the use of an ignition oven due to their good intrinsic properties (as shown in Table 1).

The data in Table 7 show that the bitumen content (AC) of all the mixes tested is lower than the mix design where the bitumen has been extracted with trichloroethylene (TCE). This difference, approximately 0.2 percent, suggests that a correction factor is necessary when using the ignition oven with this mix containing 30% RAP. The particle size distributions for most of the sieves are similar, but the 80 µm sieve shows significant variation between the mixes due to the attrition of the aggregates during the mixing process. The TR mixes have less attrition, possibly due to their lower mixing energy, while the HB and PM mixes have higher attrition and a higher content of 80 µm sieve-passing particles, possibly due to their more aggressive mixing energy. The last two mixing times (three and four) for these mixers (HB and PM) types are critical and do not meet the ESG-10 specifications. It may be beneficial to aim for a lower 80µm content when using more RAP to allow a longer mixing time.

Figure 9 shows the relation between the recorded voids and the average air voids obtained from SGC and Marshall compaction. As mixing time increases, the percent passing through the 80µm sieve also increases. This could be the reason that the voids vary for each mix, because fine particles fill the voids. On average, the voids decrease with the increase in the 80µm sieve content. Special attention to the mixing conditions in the laboratory and in asphalt plants is required to produce the right mix formula.

### 4.2. HMA Performance

#### 4.2.1. Indirect Tensile Strength Modulus (ITSM)

Table 8 shows the ITSM results at −20, −10, 0, and 10 °C. The table shows that, as expected, the ITSM results vary with temperature, with the mix becoming stiffer as the temperature decreases. The repeatability of the tests is good, as indicated by the average standard deviation of 3.4%. The maximum variation was found at 0 °C for the PM-2 mix, with 7.7%, while the lowest variation was found at −20 °C for the PM-1 mix, at 0.3%. All the results fall within the maximum difference specified by the NF EN 12697-26 standard, which is 10 percent. The average standard deviation (σ) between all mix types and time is 1319, 1308, 482, and 354 for −20, −10, 0, and 10 °C, which remained in the 10 percent maximum difference of the standard.

Overall, the stiffness of the PM, HB, and TR mixes increased as the mixing time increased. The stiffness of PM-2, PM-3, and PM-4 increased on average by 5.5%, 5.2%, and 12.1%, respectively. PM-2 and PM-3 showed similar behavior, despite having a 2.1% variance in voids. The stiffness increase in PM-4 may be due to its lower void percentage (2.0% lower than PM-1). As mentioned earlier, increasing the mixing time of the HMA affects its particle size distribution, increases binder oxidation, and activates more RAP bitumen, leading to an increase in stiffness.

The stiffness of HB-2, HB-3, and HB-4 increased with mixing time. HB-2 and HB-3 showed similar behavior with similar voids at temperatures between −10 °C and 10 °C (7.6% and 8.2% increase compared to HB-1). The lower voids in HB-4, combined with bitumen oxidation, may have contributed to its higher stiffness (10.0% increase compared to HB-1). Increasing the mixing time of the countertop mixer (HB) affects the particle size distribution of the HMA, resulting in an increase in stiffness.

The stiffness of TR-2, TR-3, and TR-4 increased on average by 3.3%, 3.9%, and 12.7%, respectively. TR-2 and TR-3 showed similar stiffness, despite having higher voids for TR-2. Mixing time appeared to have a greater effect on ITSM results at lower temperatures (−20 and −10 °C). The stiffness increase of TR-4 may be due to binder oxidation, as it was mixed for 160 s longer than TR-3 in a temperature-controlled mixer at the mixing temperature.

The one-way ANOVA conducted on the ITSM results shown in Table 9 reveals that both mixing time and mixer type have a significant effect on the means of the groups. The analysis compares all the mixers and finds that there is a significant difference in means between the groups of mixing time for all temperatures (F-value > F-crit and *p*-value < 0.05). This results in the rejection of the null hypothesis, and it demonstrates that the findings are statistically significant.

The results of the ANOVA for TR mixers indicate that there is a statistically significant difference in means between the groups for all the temperatures of −10, 0, and 10 °C. This is indicated by the F-value being greater than the F-crit value and the *p*-value being less than 0.05 (Table 9). However, at −20 °C, the F-value is lower than the F-crit value, and the *p*-value is greater than 0.05, indicating that there is not enough evidence to reject the null hypothesis, meaning that there is no significant difference in means between the groups at that temperature.

HB mixer ANOVA results indicate that there is a statistically significant difference in means between the groups for all temperatures tested. This is indicated by the F-value being greater than the F-critical value and the *p*-value being less than 0.05 for all temperatures. It suggests that mixing time does have an impact on the results for the HB mixer at any temperature.

PM mixer ANOVA results suggest that there is a statistically significant difference in means between the groups for temperatures of −10, 0, and 10 °C, as the F-value is greater than the F-crit value and the *p*-value is less than 0.05. However, at −20 °C, the F-value is lower than the F-crit value, and the *p*-value is greater than 0.05, indicating that there is not enough evidence to reject the null hypothesis, meaning that there is no significant difference in means between the groups at that temperature.

#### 4.2.2. Effect of Air Voids on ITSM Results

Figure 10 presents the ITSM results at −20, −10, 0, and 10 °C in the function of air voids. Overall, the effect of air voids on stiffness is lower than expected. For most temperatures tested, there is not a significant increase in stiffness (ISTM) with decreasing air voids (V_a_). By comparing the mix with the highest air void content (TR-1) and the lowest one (HB-4), with a difference of 4.0 percent of voids, the lowest variability is observed at 10 °C with an increase of 0.1 percent, followed by −20 °C, with a 3.2 percent increase, and 0 °C (5.9 percent increase). The effect of voids is more important on the −10 °C results, with an increase of 3086 MPa, or 15.5 percent. This could be an indication that other parameters affect the results that lower the stiffness through the mixing time. A longer mixing time could contribute to the activation of the RAP bitumen and raise the effective bitumen content. A higher bitumen content results in a lower stiffness. 

#### 4.2.3. Indirect Tensile Strength (IDT)

Figure 11 and Figure 12 show the average IDT results at 25 °C and −20 °C, respectively. The average air voids have been plotted to observe the effect of voids on indirect tensile strength. The first thing to note is that the repeatability is better at 25 °C than at −20 °C. The specimens were conditioned for 4 h at the testing temperature prior to testing. After the installation of the specimen on the MTS press, the conditioning chamber temperature changed due to the opened door, and an additional conditioning time was carried out (five minutes after reaching the testing temperature). A higher IDT value indicates a better resistance to rutting.

For HB mixes, the IDT tends to increase with the second and third mixing times but decreases compared to the first mixing time when the fourth mixing time is used at both tested temperatures (Figure 11 and Figure 12). In general, lower voids tend to result in higher IDT values, but this relationship does not hold for the HB-4 mix.

At 25 °C (Figure 11), the first mixing time produces the lowest IDT for the PM and TR mixes, and the IDT increases with each subsequent mixing time. This suggests that a longer mixing time leads to a higher quality mix for PM and TR. The highest value at 25 °C is obtained with the TR-4 mix with 918 kPa and the lowest is obtained with the PM-1 mix with 712 kPa.

At −20 °C (Figure 12), HB mixes follow the same trend as at 25 °C (Figure 11), with IDT going up until the HB-3 mixing time and then decreasing lower than HB-1. This could be related to the particle size distribution of the mix. PM-1 has the lowest IDT value, and the next mixing time is higher. The high variability limits the interpretation of the results, since PM-2, PM-3, and PM-4 are in the same range of values. They still seem to have a higher IDT value than the first mixing time. The TR mixes’ lowest IDT is obtained with the second and the third mixing time, followed by the first mixing time, and the last mixing time has the highest IDT value. Again, the high variability could affect the interpretation. The highest IDT value overall was obtained with the third mixing time of the HB mixer at 5358 kPa, while the lowest value was obtained with TR-2 at 4101 kPa.

The one-way ANOVA in Table 10 compares the IDT results of all mixers at different temperatures. At −20 °C, the analysis finds that there is no statistically significant difference in means between the groups for mixing time, as the F-value is lower than the F-crit value, and the *p*-value is greater than 0.05. However, at 25 °C, the results are statistically significant, as the F-value is higher than F-crit and the *p*-value is less than the significance level of 0.05. This suggests that mixing time has an impact on the IDT results at 25 °C but not at −20 °C when all mixer types are being compared.

The ANOVA results for the TR and HB mixers indicate that there is not enough evidence to reject the null hypothesis for both testing temperatures. This means that there is no statistically significant difference among the groups being compared in the ANOVA.

On the other hand, the PM mixer results at −20 °C suggest that mixing time has no impact on the means of the groups. However, at 25 °C, the results are statistically significant, indicating that there is a significant difference in means among the groups for mixing time. This means that the null hypothesis is rejected at 25 °C for the PM mixer, but it is not for −20 °C (F-value > F-crit and *p*-value < 0.05 in Table 10).

#### 4.2.4. Crack Tolerance Index (CT_index_)

The crack tolerance index (CT_index_) has been calculated based on the IDT data at 25 °C and −20 °C, and the results are shown in Figure 13 and Figure 14. The relationship between the CT_index_ and air voids is also shown in these figures. A higher CT_index_ value indicates greater resistance to cracking and is generally considered to be indicative of a higher quality asphalt mix. As a reference, asphalt mixes made with a non-modified bitumen typically have a CT_index_ of around 60 at 25 °C [24,25]. A lower CT_index_ value suggests that the asphalt mix is more prone to cracking under load and offers less relaxation.

At 25 °C (Figure 13), mixes demonstrate strong resistance to cracking, but the specific mixing type and duration influence the outcome. Mixes classified as HB tend to show the same trend as for IDT, but with reversed results. Compared to the first mixing time, CT_index_ decreases for the second and third mixing times but is the highest for the fourth one. This could be due to a combination of the particle size distribution and the effective bitumen content. For the PM and TR mixes, the CT_index_ is highest for the second mixing time and then decreases with longer mixing times, deviating from the trend seen in IDT mixes. The HB mixer has the biggest influence on the CT_index_ value. The highest CT_index_ value is obtained for the HB-4 mix (99) and the lowest is obtained for the HB-3 mix (58). Air voids do not seem to have any effect on the CT_index_.

As expected, at −20 °C (Figure 14), CT_index_ is a lot lower than the results at 25 °C, with an average of 0.4% of the value obtained at the previous temperature. During the test, specimens exploded when reaching their maximum value, and so did not show any crack propagation. This may be due to the fragile behavior of the material at this temperature. Overall, CT_index_ values varied from 0.17 (PM-1 and TR-2) to 0.81 (TR-4), which indicates no real resistance to cracking.

In Table 11, the one-way ANOVA reveals that there is no statistically significant difference in CT_index_ results among the mixers at various temperatures and mixing times, with the exception of the HB mixer at −20 °C, for which the F-value is greater than the F-crit value and the *p*-value is less than 0.05, indicating a significant difference.

#### 4.2.5. Work of Failure (W_f_) and Failure Energy (G_f_)

Figure 15 presents an example of IDT results for HB-1 mix at 25 °C and −20 °C. The maximum force recorded during the test was 7.32 kN at 25 °C and 46.78 kN at −20 °C. At 25 °C, the mix continues to exhibit resistance after reaching the maximum load, resulting in a higher displacement before the load drops to 0.10 kN. At −25 °C, the specimen breaks after reaching its maximum load and the test is completed more quickly. The maximum load is reached more quickly at lower temperatures.

The work of failure, or W_f_, is a measure of the energy required to cause a material to fail, and it is represented by the area under the curve in an IDT test. The failure energy, or Gf, is a measure of the amount of energy required to break a sample, considering the dimensions of the sample. Figure 16 and Figure 17 show the W_f_ and G_f_ values at 25 °C and −20 °C, respectively.

The work of failure (W_f_) and failure energy (G_f_) of asphalt tend to be higher at higher temperatures (25 °C vs −20 °C), indicating that more energy is required to cause the specimen to fail (Figure 16 and Figure 17). This is because the sample is able to store more energy at higher temperatures, even if the force required to reach the maximum load is lower. The same trend is observed for both W_f_ and G_f_, as the specimens used in the test have similar dimensions, with the exception of the air void content, which can affect the results. The mixer type and mixing time also influence the W_f_ and G_f_ values, but not significantly. Additionally, the repeatability of the test is better at a testing temperature of 25 °C.

For asphalt mixes produced using the HB mixer at a testing temperature of 25 °C, the effect of mixing time on the failure energy (G_f_) value is minimal (Figure 17). The work of failure (W_f_: Figure 16) and G_f_ values for the first mixing time are 31 J and 5006 J/m^2^, respectively, and they increase slightly for HB-2 (33 J and 5190 J/m^2^) and HB-3 (32 J and 5197 J/m^2^) but remain the same for HB-4 (31 J and 5005 J/m^2^).

For mixes produced using the PM mixer, the W_f_ and G_f_ values are lowest for the first mixing time (28 J and 4387 J/m^2^) but then stabilize at values ranging from 32 to 33 J and 4924 to 5030 J/m^2^ for the subsequent mixing times.

For mixes produced using the TR mixer, the W_f_ and G_f_ values do not follow the same trend. TR-1 has the lowest G_f_ value (5181 J/m^2^), followed by TR-3 (5224 J/m^2^) and TR-2 (5580 J/m^2^), and TR-4 has the highest G_f_ value (5833 J/m^2^). The lowest W_f_ for TR mixes is obtained with the third mixing time (TR-3, 33 J), while TR-1 and TR-2 have the same value (36 J), and TR-4 has the highest W_f_ value (37 J).

At a testing temperature of −20 °C (Figure 16 and Figure 17), the average work of failure (W_f_) and failure energy (G_f_) values are 14.0% and 14.8% lower, respectively, compared to the values obtained at 25 °C. The general trend of the W_f_ and G_f_ values at −20 °C follows the same pattern as those at 25 °C, but with higher variability due to the increased fragility of the asphalt material at lower temperatures. The average standard deviation of all the results at 25 °C is 5.9% for both W_f_ and G_f_. For −20 °C, the average standard deviation is 17.1% for W_f_ and 17.3% for G_f_, indicating a greater degree of variability in the test results at this lower temperature.

The one-way ANOVA in Table 12 compares the W_f_ and G_f_ results of all mixers at different temperatures with the mixing time.

The ANOVA of W_f_ and G_f_ results shows that at −20 °C, the F-value is lower than the F-critical value, and the *p*-value is higher than 0.05 for all the results, indicating that there is no statistically significant difference between the groups being compared. At 25 °C, when all mixer types are compared, the F-value is higher than the F-critical value, and the *p*-value is lower than 0.05, indicating that there is enough evidence to reject the null hypothesis and the results are statistically significant. The same trend is observed for TR and HB mixers, where the F-value is lower than the F-critical value and the *p*-value is higher than 0.05, indicating that the mixing time has no effect on the W_f_ and G_f_ results for those mixer types for both temperatures. However, for the PM mixer, the F-value is higher than the F-critical value, and the *p*-value is lower, indicating that mixing time has an impact on W_f_ and G_f_ results.

## 5. Conclusions

In conclusion, the study of the impact of mixing type and time on a semi-open graded hot-mix asphalt (HMA) with a high content (30%) of reclaimed asphalt pavement (RAP) was performed in the laboratory. It was shown that these factors can significantly affect the volumetric properties, compactability, and performance (mechanical properties) of the mixes. Three (3) types of mixers (thermoregulated, countertop, and pugmill mixers) and four (4) mixing times (#1 to #4) were used in the study, and a range of tests (G_mm_, ignition oven, sieve analysis, SGC, IDT, ITSM) were conducted. Overall, the results show:For volumetric properties, G_mm_ and V_a_ generally decrease with the increase in mixing time. The TR mixer reaches a point of stability after reaching a certain mixing time. On the other hand, P_ba_, VMA, V_be_, VFA, and particles that are smaller than 80µm increased with the mixing time for all mixer types, but the HB and PM mixers have a greater impact on these properties.Increasing the mixing time for all types of mixers can improve the compactability of the material, making it easier to apply on the field. The voids in the material when compacted using the Marshall method (40 blows on each side) are generally lower than those obtained using the SGC compactor at N_design_. Again, the HB and PM mixers have a greater impact on compactability.The mechanical properties of the mixes were found to be influenced more by the mixing conditions than the voids. The voids had a greater impact on the stiffness of the mix in ITSM tests at a temperature of −10 °C, but other factors, such as the mix’s volumetric properties and the mobilization of RAP bitumen, also affected the stiffness of the mix. In IDT tests, stiffness was affected by the type of mixer and mixing time, but no correlation with the voids can be observed. The CT_index_ had good results at 25 °C, but no crack resistance was observed at −20 °C due to the brittle nature of the HMA. The energy stored during the test indicated that more energy was stored at 25 °C, even though a higher maximum load was achieved at −20 °C.The ANOVA indicates that the ITSM results are the most affected by the mixing time and type. The results suggest that as the testing temperature increases, the mixing time has a more pronounced effect on ITSM and other performance parameters. This implies that at higher temperatures, the mixing time has a more significant impact on the stiffness and strength of the asphalt mixture.

The results of the study demonstrate that it is crucial to carefully consider mixing time and type to produce mixes that meet the required specifications (V_be_, V_a_, and 80µm-passing content) and have good resistance to cracking. Additionally, the impact of voids on the mixes was also examined, with the results showing that voids can have a significant impact on the properties of the mixes (V_be_, VMA, and VFA). Other factors that influence mix performance include fine particle concentration. A longer mixing time tends to generate more fine particles and facilitate the compactability of the mix. It is important to consider the generation of fine particles to ensure that the mix meets specifications. For mix design, it may be beneficial to aim for lower control points below the maximum limit after the mixing time. When using RAP—30% here—the mixing time and type of mixer used can affect the amount of RAP bitumen that is activated. This can have an impact on the performance of the mixture, so it would be valuable to conduct further research on this topic. Because of this, good control of the mix production in the laboratory is probably even more important for mixes with RAP than for virgin mixes. It would also be worthwhile to conduct further research on the impact of the mixer type and mixing time on asphalt plant mixes’ properties and performance.

## Figures and Tables

**Figure 1 materials-16-01300-f001:**
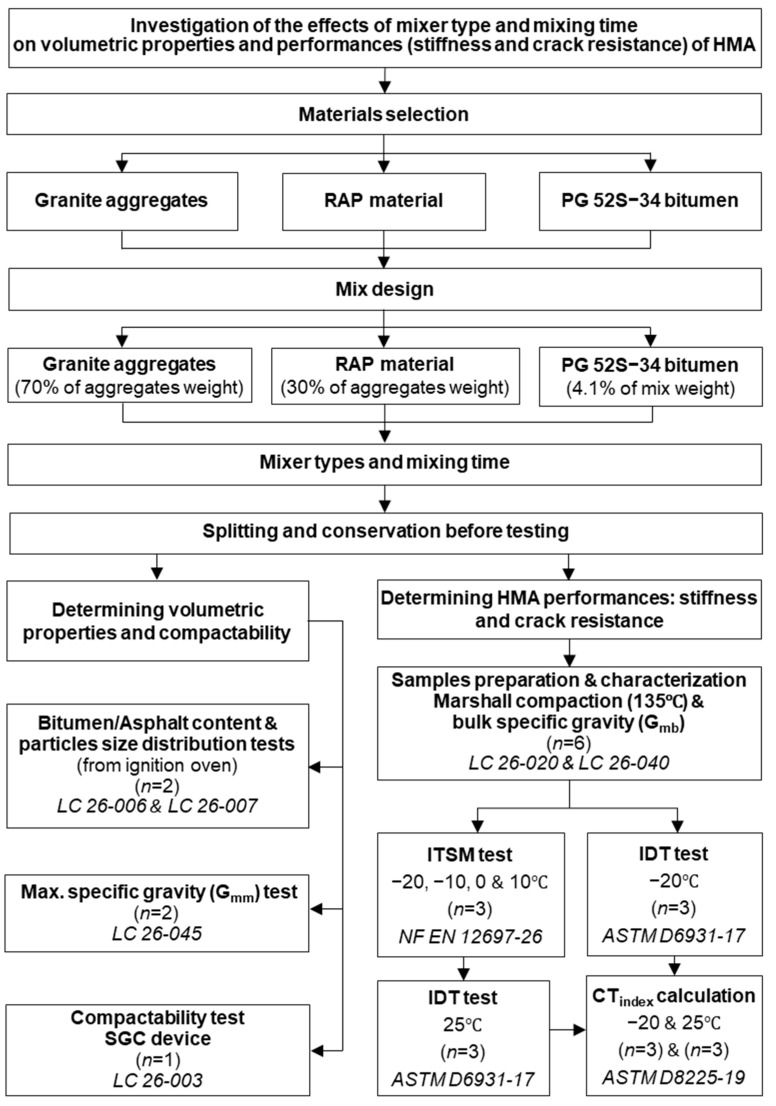
Organigram of the experimental program.

**Figure 2 materials-16-01300-f002:**
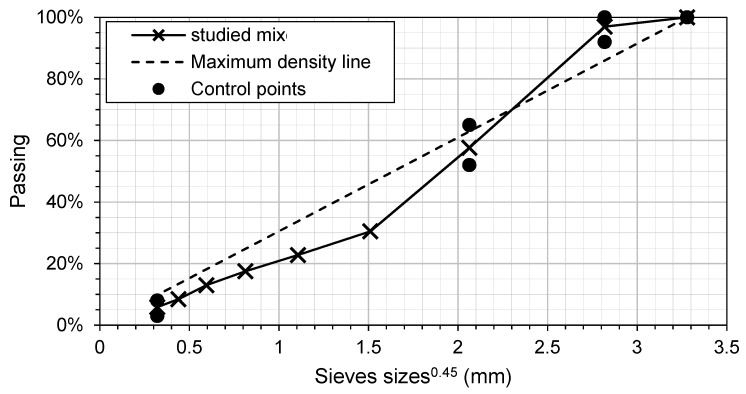
Particle size distribution of studied mix.

**Figure 3 materials-16-01300-f003:**
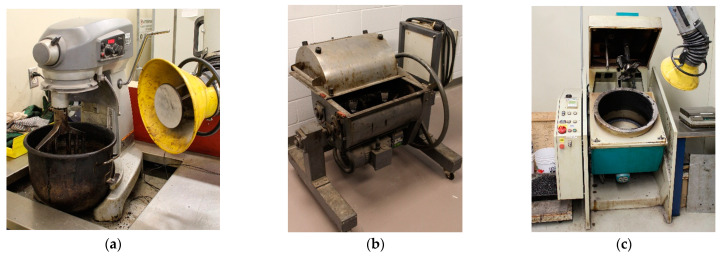
Mixers used for the study: (**a**) countertop mixer (HB); (**b**) pugmill mixer (PM); (**c**) thermoregulated mixer (TR).

**Figure 4 materials-16-01300-f004:**
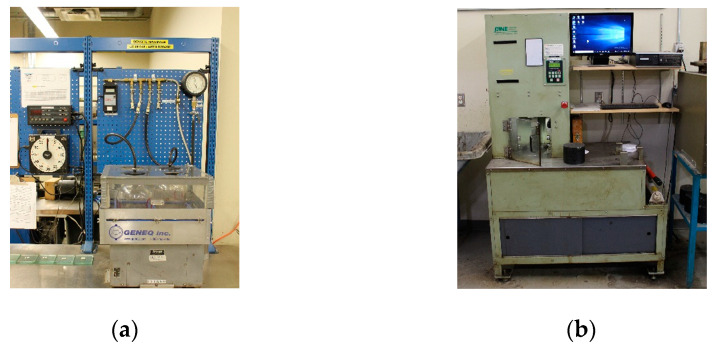
Determining HMA volumetric properties and compactability: (**a**) maximum specific gravity (G_mm_) test; (**b**) shear gyratory compactor (SGC) device.

**Figure 5 materials-16-01300-f005:**
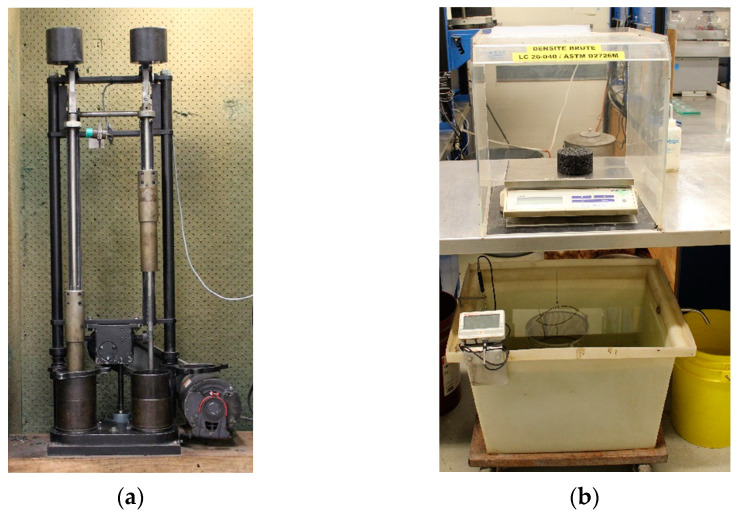
Marshall method: (**a**) compaction of the specimens; (**b**) hydrostatic weighing.

**Figure 6 materials-16-01300-f006:**
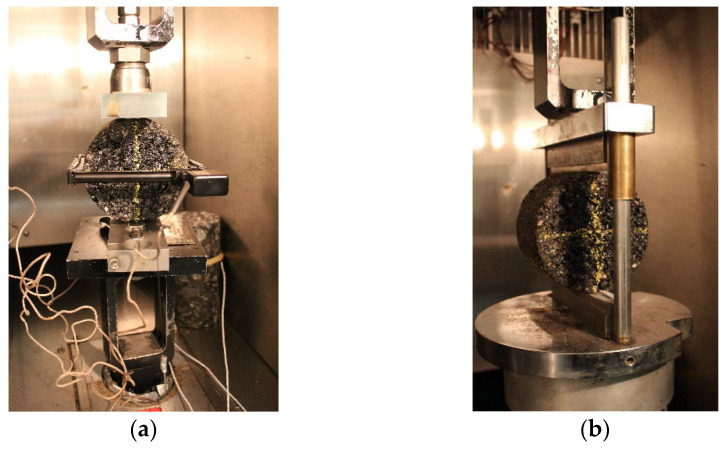
Asphalt performance evaluation: (**a**) ITSM device; (**b**) IDT device.

**Figure 7 materials-16-01300-f007:**
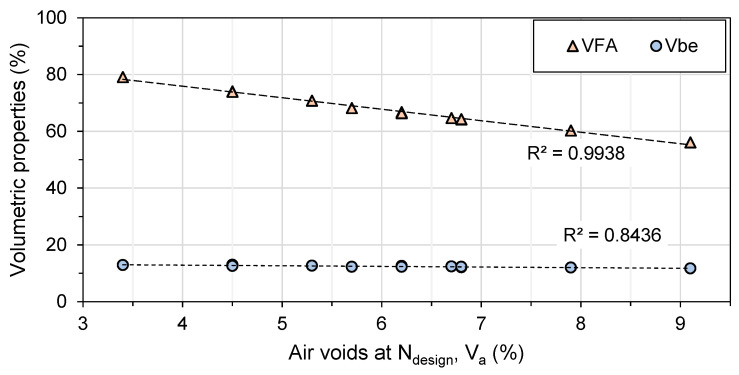
Effective volume of asphalt binder (V_be_) and voids filled with asphalt (VFA) in the function of the air voids (V_a_) at N_design_.

**Figure 8 materials-16-01300-f008:**
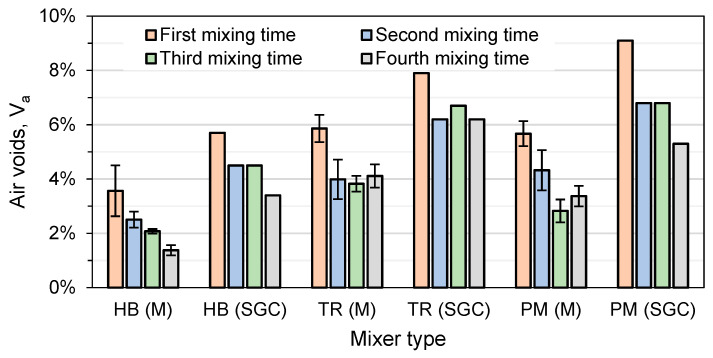
Voids with Marshall compaction (M) after 40 blows each side (*n* = 6) and shear gyratory compaction (SGC) at N_design_ (*n* = 1).

**Figure 9 materials-16-01300-f009:**
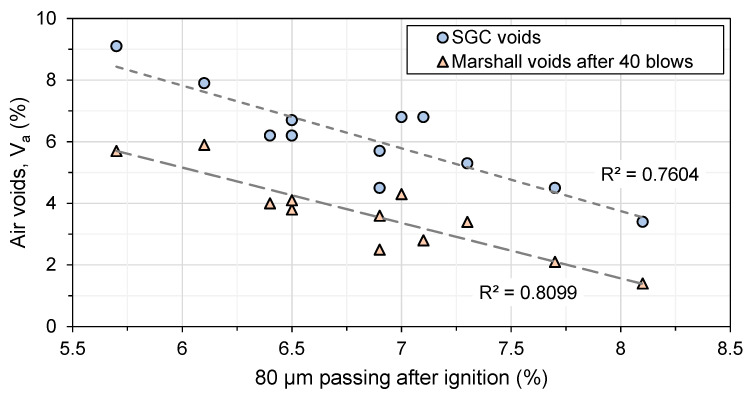
Shear gyratory compactor SGC voids (*n* = 1) and Marshall voids after 40 blows (*n* = 6) in the function of 80µm sieve-passing after mixing and an ignition test.

**Figure 10 materials-16-01300-f010:**
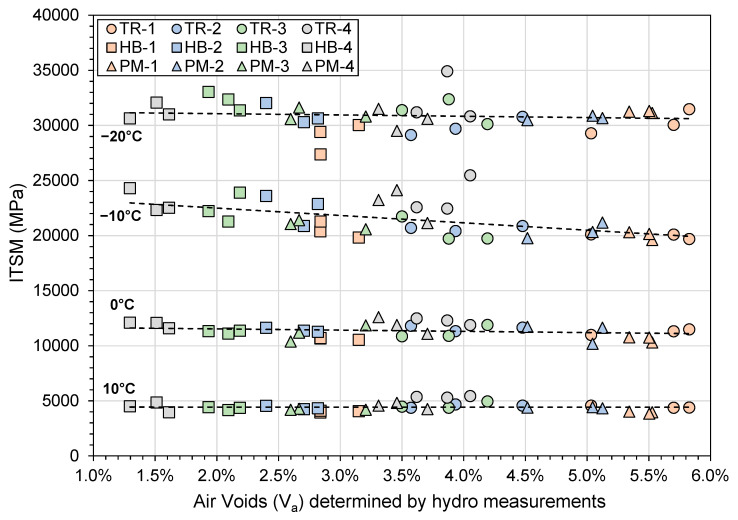
ITSM results at −20, −10, 0, and 10 °C in the function of the air voids (n = 3 for each mixing condition and thus, n = 36 for each temperature).

**Figure 11 materials-16-01300-f011:**
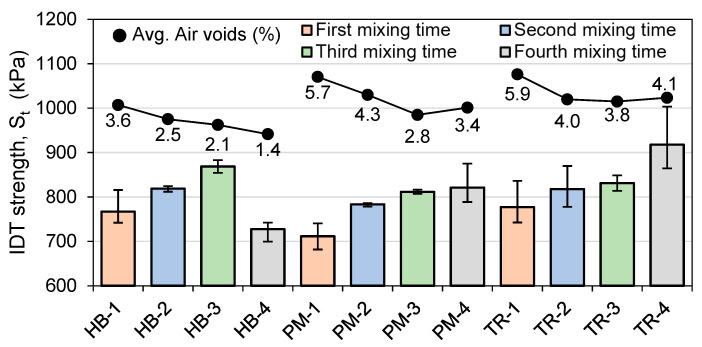
Indirect tensile strength (IDT) results at 25 °C (*n* = 3).

**Figure 12 materials-16-01300-f012:**
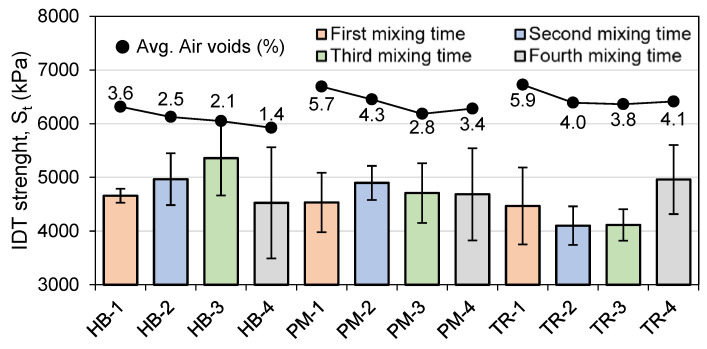
Indirect tensile strength (IDT) results at −20 °C (*n* = 3).

**Figure 13 materials-16-01300-f013:**
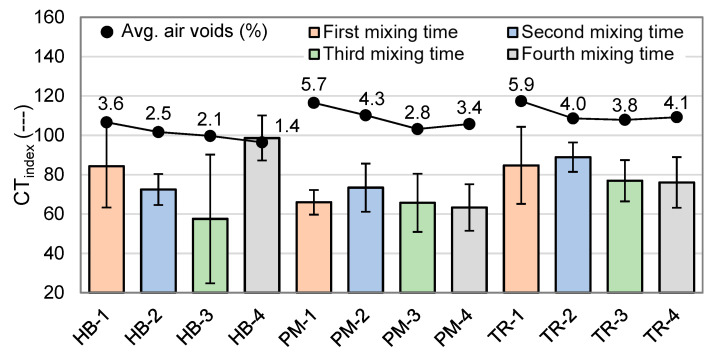
Crack tolerance index (CT_index_) results at 25 °C (*n* = 3).

**Figure 14 materials-16-01300-f014:**
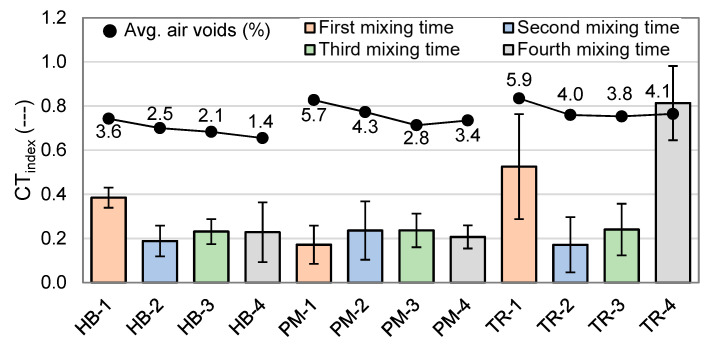
Crack tolerance index (CT_index_) results at −20 °C (*n* = 3).

**Figure 15 materials-16-01300-f015:**
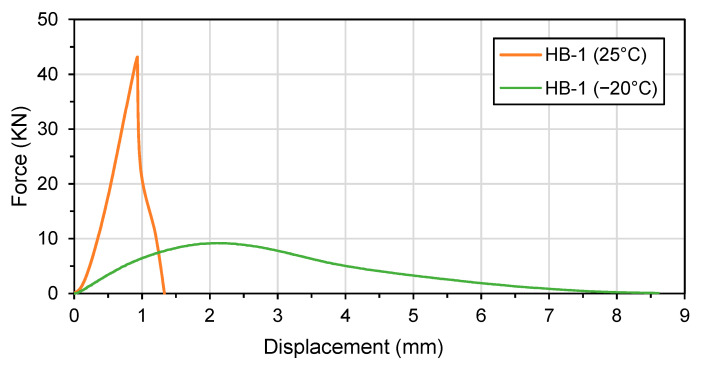
IDT results example: force recorded as a function of displacement for HB-1 specimens at 25 °C and −20 °C.

**Figure 16 materials-16-01300-f016:**
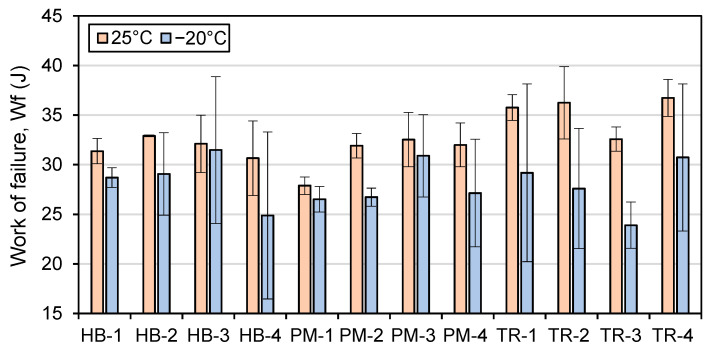
Work of failure (W_f_) results at 25 °C (*n* = 3) and −20 °C (*n* = 3).

**Figure 17 materials-16-01300-f017:**
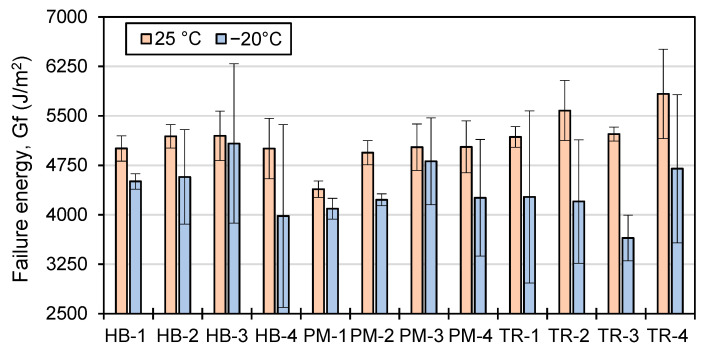
Failure energy (G_f_) results at 25 °C (*n* = 3) and −20 °C (*n* = 3).

**Table 1 materials-16-01300-t001:** Virgin aggregate properties.

Test	Units	Quebec Test Method	Results	
			Coarse Aggregates (5–10 mm)	Fine Aggregates (0–5 mm)
Micro-Deval (MD)	%	LC 21-070 and LC 21-101	9	6
Los Angeles (LA)	%	LC 21-400	27	---
Micro-Deval (MD) modif	%	LC 21-080	---	23
Crushed particles	%	LC 21-100	100	100
Flakiness index	%	LC 21-265	12.8	---
Elongation index	%	LC 21-265	33.1	---
Bulk specific gravity (G_sb_)	---	LC 21-065 and LC 21-067	2.732	2.718
Absorption (abs)	%		0.44	0.23

**Table 2 materials-16-01300-t002:** Reclaimed asphalt pavement (RAP) and virgin bitumen properties.

Test	Units	Standard	Results	
			RAP Extracted Bitumen *	Virgin Bitumen
High-temperature grading (H)	°C	AASTHO T315	84.0	52.7
Low-temperature grading (L)	°C	AASTHO T313	−24.0	−34.9
Performance grade (PG Hn-L)	---	---	PG 82-22	PG 52S-34
Mixing temperature (T_mix_)	°C	AASTHO T316	174	136
Compaction temperature (T_comp_)	°C	AASTHO T316	161	125

* Extracted with trichloroethylene (TCE).

**Table 3 materials-16-01300-t003:** Reclaimed asphalt pavement (RAP) properties.

Test	Units	Quebec Test Method	Results
Bitumen/asphalt content (AC)	%	LC 26-100	5.1
Maximum specific gravity (G_mm_)	---	LC 26-045	2.523
Water absorption (abs)	%	LC 21-065 and LC 21-067	0.0
Total specific surface area (SSA)	m^2^/kg	LC 21-040	9.97
Aggregate bulk specific gravity (G_sb_)	---	LC 21-065 and LC 21-067	2.714

**Table 4 materials-16-01300-t004:** Conditioning of materials before mixing, mixer type, and mixing time.

Conditioning before Mixing			Mixer Type	Mixing Time		Mix
Aggregates	RAP	Bitumen	Model	1st	2nd	
			(ID)	(s)	(s)	(ID)
185 °C	60 °C	145 °C	Countertop (Hobart: HB)	45	30	HB-1
					60	HB-2
					120	HB-3
					240	HB-4
185 °C	60 °C	145 °C	Pugmill (PM)	30	15	PM-1
					30	PM-2
					60	PM-3
					120	PM-4
145 °C	145 °C	145 °C	Thermoregulated (TR)	60	40	TR-1
					80	TR-2
					160	TR-3
					320	TR-4

**Table 5 materials-16-01300-t005:** Volumetric properties.

Mix (ID)	G_mm_ (*n* = 2) (---)	P_ba_ (*n* = 2)	G_mb_ ^1^ (*n* = 1)	V_a_ (*n* = 1)	V_be_ ^2^ (*n* = 2)	VMA (*n* = 2)	VFA (*n* = 2)
		(%)	(---)	(%)	(%)	(%)	(%)
TR-1	2.524	0.41	2.325	7.9	12.0	19.9	60.3
TR-2	2.514	0.24	2.357	6.2	12.6	18.8	66.8
TR-3	2.519	0.33	2.349	6.7	12.4	19.1	64.7
TR-4	2.522	0.38	2.365	6.2	12.3	18.5	66.4
HB-1	2.526	0.44	2.381	5.7	12.3	18.0	68.2
HB-2	2.509	0.13	2.395	4.5	13.0	17.5	74.1
HB-3	2.518	0.29	2.406	4.5	12.6	17.1	73.9
HB-4	2.514	0.23	2.429	3.4	12.9	16.3	79.1
PM-1	2.530	0.54	2.299	9.1	11.7	20.8	56.1
PM-2	2.525	0.39	2.353	6.8	12.1	18.9	64.2
PM-3	2.521	0.32	2.349	6.8	12.3	19.0	64.3
PM-4	2.515	0.32	2.383	5.3	12.7	18.0	70.8

^1^ G_mb_ at N_design_ (80 gyrations) from SGC results. ^2^ V_be_ value was determined with the air voids (V_a_) value at N_design_.

**Table 6 materials-16-01300-t006:** Air voids (V_a_) obtained from the SGC test.

Mix (ID)	V_a_ at Number of Gyrations (%) (*n* = 1)					
	10	60	80	100	120	200
TR-1	16.5	9.0	**7.9**	7.2	6.6	5.0
TR-2	15.1	7.4	6.2	5.5	4.8	3.4
TR-3	15.5	7.8	6.7	5.9	5.4	3.8
TR-4	15.1	7.3	6.2	5.4	4.8	3.4
HB-1	14.4	6.8	5.7	4.9	4.4	2.9
HB-2	13.5	5.6	4.5	3.7	3.1	**1.7**
HB-3	13.5	5.5	4.5	3.7	3.1	**1.8**
HB-4	12.4	4.4	**3.4**	2.7	2.2	**1.2**
PM-1	17.6	10.2	**9.1**	8.4	7.8	6.3
PM-2	15.6	7.9	6.8	6.0	5.4	3.9
PM-3	15.5	7.8	6.8	6.0	5.4	3.9
PM-4	14.1	6.3	5.3	4.5	3.9	2.5
**N_name_** **Spec.**	N_ini_ ≥11.0	--- ---	N_design_ 7.0–4.0	--- ---	--- ---	N_max_ ≥2.0

**Table 7 materials-16-01300-t007:** Bitumen content and particle size distribution from ignition.

Mix (ID)	AC ^1^ (%) (*n* = 2)	Particle Size Distribution (*n* = 2)								
		(mm)					(µm)			
		14	10	5	2.5	1.25	630	315	160	80
TR-1	5.37	100	97	57	30	21	17	13	9	6.1
TR-2	5.45	100	97	56	30	21	17	13	9	6.4
TR-3	5.41	100	96	57	30	21	17	13	9	6.5
TR-4	5.42	100	97	57	29	21	17	13	9	6.5
HB-1	5.50	100	96	58	31	22	18	13	10	6.9
HB-2	5.44	100	97	57	31	22	17	13	10	6.9
HB-3	5.48	100	97	58	32	23	19	15	11	7.7
HB-4	5.42	100	97	57	31	22	18	14	11	8.1
PM-1	5.45	100	96	53	27	19	15	11	8	5.7
PM-2	5.41	100	96	56	30	22	18	13	10	7.0
PM-3	5.42	100	96	58	31	23	18	14	10	7.1
PM-4	5.44	100	97	58	32	23	18	14	10	7.3
Ref.	5.60	100	97	58	30	23	18	13	8	5.8
Spec.	^---^	100	100	65	46.1 ^2^	36.7 ^2^	26.8 ^2^	18.1 ^2^	---	7
		100	92	52	46.1 ^2^	30.7 ^2^	22.8 ^2^	18.1 ^2^	---	4

^1^ Asphalt content based on V_be_ = 12.2%. ^2^ It is recommended to pass outside of this restrictive area.

**Table 8 materials-16-01300-t008:** ITSM results at −20, −10, 0, and 10 °C.

Mix		Testing Temperature (°C)							
		−20 (*n* = 3)		−10 (*n* = 3)		0 (*n* = 3)		10 (*n* = 3)	
		E′	σ	E′	σ	E′	σ	E′	σ
ITSM (MPa)	PM-1	30,821	92	20,009	372	10,598	270	3935	91
	PM-2	31,883	197	20,421	713	11,180	864	4374	51
	PM-3	32,000	546	21,004	418	11,139	730	4215	53
	PM-4	33,000	998	22,830	1531	11,846	757	4539	287
	HB-1	29,765	1399	20,489	736	10,630	91	4030	6
	HB-2	31,221	920	22,444	1410	11,419	188	4368	162
	HB-3	32,827	846	22,456	1326	11,260	140	4313	142
	HB-4	31,397	957	23,046	1211	11,915	94	4439	86
	TR-1	28,576	1113	19,959	236	11,255	255	4436	106
	TR-2	29,887	834	20,655	227	11,585	247	4528	155
	TR-3	31,562	1130	20,395	115	11,212	585	4592	308
	TR-4	32,304	2262	23,493	1708	12,211	310	5356	72

**Table 9 materials-16-01300-t009:** One-way ANOVA of ITSM results at −20, −10, 0, and 10 °C (significance level = 0.05).

	−20 °C			−10 °C		
	F-Value	*p*-Value	F-Crit	F-Value	*p*-Value	F-Crit
All mixer	3.159	0.002	1.952	8.607	0.000	1.952
TR	2.290	0.109	3.098	14.840	0.000	3.098
HB	9.892	0.000	3.098	6.126	0.004	3.098
PM	0.825	0.495	3.098	7.024	0.002	3.098
**Mixer type**	**0 °C**			**10 °C**		
	**F-value**	***p*-value**	**F-crit**	**F-value**	***p*-value**	**F-crit**
All mixers	6.161	0.000	1.952	19.221	0.000	1.952
TR	8.319	0.001	3.098	8.319	0.000	3.098
HB	14.138	0.000	3.098	3.242	0.044	3.098
PM	3.844	0.025	3.098	16.007	0.000	3.098

**Table 10 materials-16-01300-t010:** One-way ANOVA of IDT results at −20 and 25 °C (significance level = 0.05).

	−20 °C			25 °C		
	F-Value	*p*-Value	F-Crit	F-Value	*p*-Value	F-Crit
All mixers	0.693	0.733	2.216	4.100	0.002	2.216
TR	0.755	0.550	4.066	2.120	0.176	4.066
HB	0.826	0.516	4.066	0.468	0.713	4.066
PM	1.043	0.425	4.066	5.037	0.030	4.066

**Table 11 materials-16-01300-t011:** One-way ANOVA of CT_index_ results at −20 and 25 °C (significance level = 0.05).

	−20 °C			25 °C		
	F-Value	*p*-Value	F-Crit	F-Value	*p*-Value	F-Crit
All mixer	1.038	0.446	2.216	1.405	0.234	2.216
TR	0.863	0.499	4.066	1.078	0.412	4.066
HB	4.667	0.036	4.066	0.946	0.463	4.066
PM	0.338	0.799	4.066	0.525	0.678	4.066

**Table 12 materials-16-01300-t012:** One-way ANOVA of W_f_ and G_f_ results at −20 and 25 °C (significance level = 0.05).

	Mixer Type	−20 °C			25 °C		
		F-Value	*p*-Value	F-Crit	F-Value	*p*-Value	F-Crit
W_f_	All mixers	0.693	0.733	2.216	4.100	0.002	2.216
	TR	0.755	0.550	4.066	2.120	0.176	4.066
	HB	0.826	0.516	4.066	0.468	0.713	4.066
	PM	1.043	0.425	4.066	5.037	0.030	4.066
G_f_	All mixers	0.756	0.678	2.216	3.195	0.008	2.216
	TR	0.717	0.569	4.066	1.637	0.256	4.066
	HB	0.823	0.517	4.066	0.336	0.800	4.066
	PM	0.971	0.453	4.066	4.297	0.044	4.066

## Data Availability

The data presented in this study are available on request from the corresponding author.

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
