# Peer review of "Laboratory Study of the Effects of the Mixer Type and Mixing Time on the Volumetric Properties and Performance of a HMA with 30 Percent Reclaimed Asphalt Pavement"

_materials, 2023, doi:10.3390/ma16031300_

Round 1
Reviewer 1 Report
1. In 3.4 section, all components were at the Tmix (145°C). Please explain the meaning of Tmix145°C? What is the applicable specification for this temperature? Similarity, please explain the temperature of conditioning aggregates were 145°C and 185°C.
2. In 3.7 section, a second compaction method was performed using the Marshall hammer (40 blows per side and six (6) specimens per mix). What is the applicable specification for 40 blow?
3. When using RAP, 30% here, the mixing time and the mixer type can have an influence on the amount of the RAP bitumen that is activated. Base on, does this result mean that the RAP usage should not exceed 30%? If the voids is well controlled, can the RAP usage be increased?
Author Response
Thank you for the revision.
- In 3.4 section, all components were at the Tmix (145°C). Please explain the meaning of Tmix145°C? What is the applicable specification for this temperature? Similarity, please explain the temperature of conditioning aggregates were 145°C and 185°C.
Tmix and Tcomp are defined on section 3.1. Precisions have been made for conditioning.
- In 3.7 section, a second compaction method was performed using the Marshall hammer (40 blows per side and six (6) specimens per mix). What is the applicable specification for 40 blows?
We wanted higher voids in sample (similar to SCG sample). Even by doing that, we obtained lower voids. info added in the paper
- When using RAP, 30% here, the mixing time and the mixer type can have an influence on the amount of the RAP bitumen that is activated. Base on, does this result mean that the RAP usage should not exceed 30%? If the voids is well controlled, can the RAP usage be increased?
The next paper will be about the RAP bitumen activation with mixing time. A precision has been added in the conclusion.
Reviewer 2 Report
The research described is very interesting. It describes and finds significant results both from a scientific point of view and from a technical and practical point of view.
In my point of view, the paper is well-written and well-organized. Here are some suggestions to add and improve the overall text and research exposition:
1) background: add some research describing the same topic investigated in the paper. If these do not exist (it seems strange to me) specify them well in the text and base the objective and research motivation on this point.
2)3.3) Mixer Types and Mixing Times
Here describe more of the different mixing processes. Add some motivations (technical) for the different mixing times selected. This is the focus of the research and needs to be well described.
3) results: the results find are statistically significant?? No tests were performed. In my point of view, doing some simple statistical tests and finding the statistical significance would make the research conducted stronger.
4) conclusion: add some observations (more technical) that allow the practitioners to improve the quality of the HMA mixture.
Author Response
The research described is very interesting. It describes and finds significant results both from a scientific point of view and from a technical and practical point of view. In my point of view, the paper is well-written and well-organized. Here are some suggestions to add and improve the overall text and research exposition:
Thank you for the comments on our paper. We did all the corrections proposed by the reviewers.
- background:
Add some research describing the same topic investigated in the paper. If these do not exist (it seems strange to me) specify them well in the text and base the objective and research motivation on this point.
A section has been added about the inclusion on RAP material and the impact of the behavior on HMA.
“Studies on laboratory-produced mixtures have found that adding low amounts of RAP (15-20%) does not significantly affect the stiffness and strength of the mix at different temperatures [6, 7, 8]. However, increasing the RAP content beyond 20% improves the mixture's stiffness and strength, resulting in increased resistance to rutting [9]. For higher RAP contents (>40%), the stiffness of the mix increases significantly at various temperatures when no changes to the virgin binder grade are made [10, 11]. The majority of Canadian agencies have established a range for the maximum RAP content allowed in new mixes, with the typical range being between 20% to 50%. The specific maximum RAP content in new mixes can vary depending on the province or territory, but on average, the RAP content used in new mixes is around 10% [12].”
- (2.3.3) Mixer Types and Mixing Times:
Here describe more of the different mixing processes. Add some motivations (technical) for the different mixing times selected. This is the focus of the research and needs to be well described.
Precision about the models used has been provided.
- results:
The results find are statistically significant?? No tests were performed. In my point of view, doing some simple statistical tests and finding the statistical significance would make the research conducted stronger.
Thank you for the comment. We agree and a one-way ANOVA tests have been performed to assess the impact of the mixing time on the performance tests results (Section 4.2.1.).
- conclusion:
Add some observations (more technical) that allow the practitioners to improve the quality of the HMA mixture.
Precision has been added for designer: aim for the lower control point in gradation to respect the specification.
Reviewer 3 Report
This paper presents a study on investigation of the effect of mixing time and mixer type on the asphalt mixtures containing 30% RAP in the laboratory. The experiments are adequate and the results may be of benefits to improve laboratory preparation and mixing practices of asphalt mixtures, however, there are many places that require authors to consider improving this paper. Specific comments are provided below.
L25: recycled asphalt pavement is used herein, while reclaimed asphalt pavement is used in title. Please be consistent throughout the paper.
L33: The sentence that “However, it is important to carefully consider the potential effects of using RAP before incorporating it into an asphalt mix …” is a redundancy, as there is a sentence in L27 that “However, it is important to understand how RAP can affect the properties and performances of the resulting mix.”
Literature review is limited. There are many studies on the RAP. The authors should include those literatures relevant to your study that uses 30% RAP, and conduct a more in-depth review.
L63 to L67: The review on the mixing time and the type of mixer are limited. More exploration should be conducted.
L68 to L74: The justification on why this study on the 30% RAP asphalt mixture is needed is missing. The objectives are not well described.
Where does the type of mixer refer to, in laboratory or in asphalt plant?
Figure 1 is an experimental program, as the authors stated. But why the figure contains sections, which makes it look like a structure of thesis or report? This flow chart needs a revision to be more specific and logic.
L92: Do not repeat spelling out RAP. Only spell it out the first time it appears in the text. Please check it throughout this paper, for examples, L104, L113, and many other places.
Table 2: Why is fine aggregates ranging from 2.5 to 5 mm? How to measure the crushed particles for fine aggregates? The criteria for each parameter, if any, is missing.
What type of aggregate, limestone, granite, or? What is the gradation of blended aggregate?
L117: The RAP is dried in an oven at 40C for how long?
Section 3.3: Details for each mixer type should be provided, such as technical parameters, temperature controlling, torque, cost, etc.
The authors only introduced the mixing. How about compaction? It appears that the authors introduced compaction using Marshall Hammer method in Section 3.7, but also mentioned SGC in Section 3.6. Why are there two different compaction methods in this study?
Section 3.7: Are all specimens subject to short-term aging or long-term aging? Please describe.
What is the designed air voids content?
Table 5, Volumetric Properties: How many replicates? Please show the standard deviation and conduct a statistical analysis.
L234: Please cite relevant reference to support the claim that “In principle, higher absorption leads to a higher Gmm value”.
Table 6, Air Voids: How many replicates? Please show the standard deviation and conduct a statistical analysis.
Figure 7 can be deleted, as you have Table 5.
Place a horizontal line in Figure 8 to show the designed air voids content. Also, please try to include air voids contents from both Marshall and SGC in one figure, for a better comparison.
Table 7: bitumen content result is confusing. Why is it necessary to measure bitumen content? Aren’t all asphalt mixtures to be designed to have the same asphalt content?
Section 4.1.2 appears to deviate from this study. Please justify how this result of bitumen content and gradation is helpful to understand the effect of mixing time and type of mixer.
Statistical analysis should be conducted for Table 8 ITSM results, Figure 11 of IDT results, Figure 12, Figure 13, Figure 14, Figure 16, and Figure 17.
Many figures miss legends for each color, such as Figures 11, 12, 13, and 14.
The authors should be cautious about the analysis techniques utilized in this study. It is highly recommended to conduct statistical analysis to improve the rigor of this study; otherwise, the conclusions made are not well justifiable.
In addition, the recommendation should be provided for the best practices of laboratory preparation of asphalt mixtures.
Author Response
This paper presents a study on investigation of the effect of mixing time and mixer type on the asphalt mixtures containing 30% RAP in the laboratory. The experiments are adequate and the results may be of benefits to improve laboratory preparation and mixing practices of asphalt mixtures, however, there are many places that require authors to consider improving this paper. Specific comments are provided below.
Thank you for the comments, we did address all of them.
L25: recycled asphalt pavement is used herein, while reclaimed asphalt pavement is used in title. Please be consistent throughout the paper.
Every recycled asphalt pavement has been replaced for reclaimed asphalt pavement.
L33: The sentence that “However, it is important to carefully consider the potential effects of using RAP before incorporating it into an asphalt mix …” is a redundancy, as there is a sentence in L27 that “However, it is important to understand how RAP can affect the properties and performances of the resulting mix.”
L33 has been reformulate with more precision. L27 is more about RAP properties and L33 is about the impact of the RAP in HMA mixes.
L27: However, it is important to understand how RAP can affect the properties and performances of the resulting mix.
L33: Despite those potential benefits of RAP, it is crucial to carefully evaluate the potential effects of adding RAP before including it in the asphalt mix, particularly its performance at low temperatures.
Literature review is limited. There are many studies on the RAP. The authors should include those literatures relevant to your study that uses 30% RAP and conduct a more in-depth review.
A short section has been added about the inclusion on RAP material and the impact of the behavior on HMA.
“Studies on laboratory-produced mixtures have found that adding low amounts of RAP (15-20%) does not significantly affect the stiffness and strength of the mix at different temperatures [6, 7, 8]. However, increasing the RAP content beyond 20% improves the mixture's stiffness and strength, resulting in increased resistance to rutting [9]. For higher RAP contents (>40%), the stiffness of the mix increases significantly at various temperatures when no changes to the virgin binder grade are made [10, 11]. The majority of Canadian agencies have established a range for the maximum RAP content allowed in new mixes, with the typical range being between 20% to 50%. The specific maximum RAP content for RAP in new mixes can vary depending on the province or territory, but on average, the RAP content used in new mixes is around 10% [12].”
L63 to L67: The review on the mixing time and the type of mixer are limited. More exploration should be conducted.
As mentioned in the text, there is little information available in the literature on the impact of mixer type and mixing time. However, a sentence was added at the end of the paragraph: “It could however be mentioned that higher mixing time or higher mixing energy could allow better reactivation of the RAP bitumen, which would results in different properties of the mixes.”
L68 to L74: The justification on why this study on the 30% RAP asphalt mixture is needed is missing. The objectives are not well described.
Precision has been added.
Where does the type of mixer refer to, in laboratory or in asphalt plant?
This mix was done in laboratory but there is ongoing work to compare the performance/properties with the same materials but produced in an asphalt plant. A precision has been added in the conclusion.
Figure 1 is an experimental program, as the authors stated. But why the figure contains sections, which makes it look like a structure of thesis or report? This flow chart needs a revision to be more specific and logic.
The figure was modified
L92: Do not repeat spelling out RAP. Only spell it out the first time it appears in the text. Please check it throughout this paper, for examples, L104, L113, and many other places.
Done for L104, L328, L525. 1 spelling has been left in the abstract, the introduction and the conclusion.
Table 2: Why is fine aggregates ranging from 2.5 to 5 mm? How to measure the crushed particles for fine aggregates? The criteria for each parameter, if any, is missing.
It was a mistake, it is for 0-5mm aggregate.
What type of aggregate, limestone, granite, or? What is the gradation of blended aggregate?
L99: crushed granite virgin aggregates.
Figure 2 shows the gradation of the blended aggregate (combined with RAP considering the white curve of the material). We do not have the information about the type of aggregate of the RAP.
L117: The RAP is dried in an oven at 40C for how long?
A precision has been added (3 days).
Section 3.3: Details for each mixer type should be provided, such as technical parameters, temperature controlling, torque, cost, etc.
Precision about the model used has been provide. With this, the reader can find the complete specs of the mixers used. We believe that adding all the specs of the mixers here would take too much space.
The authors only introduced the mixing. How about compaction? It appears that the authors introduced compaction using Marshall Hammer method in Section 3.7, but also mentioned SGC in Section 3.6. Why are there two different compaction methods in this study?
In order to follow the requirements of the Quebec method for mix design, the SGC test is used, and the Marshall compactor is employed to create samples for IDT and ITSM tests. A sub objective has been added.
Section 3.7: Are all specimens subject to short-term aging or long-term aging? Please describe.
No aging has been done on the specimens. A precision has been added.
What is the designed air voids content?
For the LC Quebec method, there is no designed air voids content. It is with the Vbe (12.2% for this type of mix), which is explain on L255. A precision is added in the methodology. With the design Vbe, we need to be between 4 and 7% at Ndesign.
Table 5, Volumetric Properties: How many replicates? Please show the standard deviation and conduct a statistical analysis.
The number of replicas is shown under the properties (n=xx).
L234: Please cite relevant reference to support the claim that “In principle, higher absorption leads to a higher Gmm value”.
A reference has been added ([9]). When doing the calculation, we see that Gmm goes up when Pba goes down.
Table 6, Air Voids: How many replicates? Please show the standard deviation and conduct a statistical analysis.
Air voids is obtained from the SCG results, where only one test has been done. It is not possible to show the standard deviation and doing a statistical analysis.
Figure 7 can be deleted, as you have Table 5.
We believe Fig 7 is a nice representation, so we prefer to keep it there even if it seems redundant
Place a horizontal line in Figure 8 to show the designed air voids content. Also, please try to include air voids contents from both Marshall and SGC in one figure, for a better comparison.
Figure 8 already presents air voids of SGC and Marshall. The designed air voids content does not exist for LC mix design method, but it must be between 4 and 7% for SCG compaction as shown on Table 6. There is no specification for Marshall compaction method for this type of mix.
Table 7: bitumen content result is confusing. Why is it necessary to measure bitumen content? Aren’t all asphalt mixtures to be designed to have the same asphalt content?
It confirms that a correction factor is needed to obtain the real bitumen content. The low variability between mixes shows that they have the same bitumen content (around 0.2% lower than what we put in it).
Section 4.1.2 appears to deviate from this study. Please justify how this result of bitumen content and gradation is helpful to understand the effect of mixing time and type of mixer.
It shows that when you mix longer, fine particles are generated. It can be critical to meet the gradation specification.
Statistical analysis should be conducted for Table 8 ITSM results, Figure 11 of IDT results, Figure 12, Figure 13, Figure 14, Figure 16, and Figure 17.
ANOVA statistical analysis have been added for those results.
Many figures miss legends for each color, such as Figures 11, 12, 13, and 14.
Legends have been added for all the Figures.
The authors should be cautious about the analysis techniques utilized in this study. It is highly recommended to conduct statistical analysis to improve the rigor of this study; otherwise, the conclusions made are not well justifiable.
Thank you for the recommendation, the conclusion has been reinforced with the statistical analyses
In addition, the recommendation should be provided for the best practices of laboratory preparation of asphalt mixtures.
The conclusion was slightly modified, but more work is needed to provide a complete best practice statement.
Round 2
Reviewer 3 Report
Comments are addressed. It is highly recommended that the authors further proofread the paper.